



# Linking compound weather extremes to Mediterranean cyclones, fronts and air streams

Alice Portal[1], Shira Raveh-Rubin[2], Jennifer L Catto[3], Yonatan Givon[2], and Olivia Martius[1]

[1]Institute of Geography, Oeschger Centre for Climate Change Research, University of Bern, Bern, Switzerland
[2]Department of Earth and Planetary Sciences, Weizmann Institute of Science, Rehovot, Israel
[3]Department of Mathematics and Statistics, University of Exeter, Exeter, United Kingdom

**Correspondence:** Alice Portal (alice.portal@unibe.ch)

**Abstract.** Mediterranean cyclones are the primary driver of many types of surface weather extremes in the Mediterranean region, the association with extreme rainfall being the most established. Although smaller in size compared to Atlantic cyclones, they share a similar synoptic structure organised in distinct air streams, such as the warm conveyor belt and the dry intrusion, and are associated with low-level temperature fronts. The large-scale characteristics of a Mediterranean cyclone, the properties of the associated airflows, the interaction with the topography around the Mediterranean basin, and the season of occurrence, all contribute in determining its surface impacts. Here, we take these factors into account to establish statistical links between Mediterranean cyclones and weather compounds of two types, namely co-occurring rain–wind and wave–wind extremes. Specifically, compound extremes are attributed to a cyclone if they fall within the system's impact area, using a definition that is expressly tested on Mediterranean cyclones and on the compound selection. Our results show that the majority of Mediterranean compound rain–wind and wave–wind extremes occur in the neighbourhood of a Mediterranean cyclone, with peaks exceeding 80%; the proportion of cyclone-related compounds is highest when considering transition seasons, and rain–wind events. Winter cyclones show highest compound frequency, matching with the peak winter occurrence of distinctively baroclinic cyclones. A novelty of this work, the de-construction of the cyclones' impact areas based on the presence of objectively-identified air streams and fronts, reveals a high incidence of both types of compound extremes below warm conveyor belt ascent regions, of wave–wind extremes below dry intrusions.

## 1 Introduction

In November 2011 a Mediterranean cyclone named Rolf struck the western Mediterranean basin causing fatalities and damage in the Balearic Islands, Southern France, Corsica and Northern Italy. The cyclone formed on 5th November through the interaction of a strong eastward-propagating front with the Pyrenees; it then remained stationary south of the French coast for approximately four days, where it developed tropical like characteristics (see details in Dafis et al., 2018), before making landfall in south-eastern France and dissipating on 10th November. The stationarity and intensity of Rolf resulted in persistent torrential rainfall, strong winds and high waves, leading to floods, landslides and treefalls (Wikipedia, 2013). Major impacts, ranging from house evacuations, traffic disruptions and blackouts to costly infrastructural damage in the public and private sectors, affected "Var département" in France and "regione Liguria" in Italy, extending to other adjacent areas (Impact Forecasting



LLC, 2011). This is only one example of high-impact compound extreme events brought by the passage of a Mediterranean cyclone.

    The compounding of extreme events (referred in the following as "compound extremes" or simply "compounds") constitutes a higher societal and economic risk compared to single extreme events (Zscheischler et al., 2018; Seneviratne et al., 2012; Ridder et al., 2020; Zscheischler et al., 2020). We focus on the co-occurrence of meteorological hazards, specifically of extreme

rain[1] and wind, and extreme wave-and-swell-height and wind, because of the potential impacts of these weather conditions in the European region (Ridder et al., 2020).

    Severe weather is often associated with the presence of extratropical cyclones (e.g., Jansa et al., 2001; Nissen et al., 2010), or with distinct cyclone features (air streams and fronts) (e.g., Madonna et al., 2014; Raveh-Rubin, 2017; Catto and Pfahl, 2013). Coherent air streams and surface impacts, arranged around the storm's low-pressure center by the anticyclonic circulation, are

described in the conceptual extratropical-cyclone model (e.g., Carlson, 1980; Schultz et al., 2019, and references therein). In the eastern warm sector, warm moist air, namely the *warm conveyor belt*, ascends northward and around the cyclone center, producing intense precipitation. The *cold front*, separating the warm and cold sector of the cyclone, trails to the south of the storm and produces pre-frontal convective activity. In satellite imagery the combination of warm conveyor belt and cold front form a synoptic-scale comma shaped cloud structure wrapping around the cyclone from the south east. In the western cold

sector, anomalously dry and cold airflows descending from the north generate a predominantly cloud-free region, characterised by isolated convection forming cumulus-type clouds. Specifically, *dry intrusions* transport cold and dry air from the upper troposphere and, when interacting with the warmer boundary layer, may generate strong surface wind gusts and low-level instability to the south of the cyclone (Raveh-Rubin, 2017).

    Mediterranean cyclones share the aforementioned dynamical features (Flaounas et al., 2022, 2015), although they are gen-

erally shorter-lived and smaller than their Atlantic equivalents (e.g. Campins et al., 2011; Lionello et al., 2016), and their interactions with regional topography may modify the distribution of air streams and surface impacts. A recent classification of Mediterranean cyclones based on their large scale dynamics unveils differences in terms of life-cycle characteristics, spatio-temporal occurrence and associated meteorological hazards (Givon et al., 2023), and proves useful for understanding which types of cyclones are most prone to weather compounding.

The passage of cyclones over or around the Mediterranean basin is statistically well-linked to regional (extreme) weather conditions in terms of precipitation (Pfahl and Wernli, 2012; Jansa et al., 2001; Pfahl, 2014) and wind (Pfahl, 2014; Nissen et al., 2010; Dowdy and Catto, 2017). The cases of rain–wind and wave–wind compounds and of their relation to extratropical cyclones have received attention on larger objective regions (Owen et al., 2021; Ridder et al., 2020; Catto and Dowdy, 2021); in the Mediterranean area, previous analyses have focused on a specific spatial scale of rain–wind events (Raveh-Rubin and

Wernli, 2015) or on a Lagrangian cyclone-centred perspective (Rousseau-Rizzi et al., 2023). The concomitant occurrence of high waves and strong winds within the Mediterranean Sea has been poorly studied, in spite of the considerable impacts, e.g., infrastructural damage along the coast, interruption of vessel traffic, or ship accidents (Cavaleri et al., 2012; Bertotti and Cavaleri, 2008; Zhang and Li, 2017) - in particular with many people crossing the Mediterranean on poor quality boats.

---

[1]The term rain is here generalised to indicate precipitation





In this work, we benchmark analyses of the distributions of compound extremes (rain–wind and wave–wind, Section 2.1) and of their association with Mediterranean cyclone tracks (Section 2.2) against existing literature. Our work extends previous studies by using the latest datasets to examine a range of yet unexplored settings and questions. In particular, we analyse mutual dependencies between compound extremes and Mediterranean cyclones to tackle the following research questions:

1. Does the presence of a cyclone impact the frequency of compound extremes? What proportion of compounds is associated with a nearby cyclone? (Sections 4, 4.1, 4.2)

2. Which amongst cyclone cold fronts, dry intrusions or warm conveyor belts (referred to collectively as *dynamical features*) is the most important for compounding extremes? (Sections 4.1, 4.2)

3. Extending the work by Givon et al. (2023), what types of cyclones are locally relevant for the occurrence of compound extremes? (Section 4.3)

While exploring the statistics of compounds and cyclones from a (geographical) Eulerian perspective, we put equivalent emphasis on winter and on transition seasons (autumn, spring) because of severe cyclone-related weather conditions occurring at such times.

An additional yet fundamental methodological question concerns the attribution of weather events to Mediterranean cyclones; this has been done in Flaounas et al. (2018) for the case of intense regional rainfall, but their method is too restrictive for identifying extreme surface winds and waves, often happening at some distance from the storm centre (Nissen et al., 2010; Pfahl, 2014; Raveh-Rubin and Wernli, 2016). Alternative reference approaches are designed for extratropical cyclones in general, which normally develop over open ocean (e.g., Catto and Dowdy, 2021). In order to assess the link between the compounding of extreme weather conditions and Mediterranean cyclones, we propose an *impact area* that, updated at every instance of the cyclone trajectory, takes into account the type of compounds, the peculiarity of the geographic setting and the synoptic air streams and fronts developing around the system (Section 3).

## 2 Data and methods

### 2.1 Compound extremes

Uni-variate extremes of 6 h cumulative precipitation (rain), maximum 10 m wind gust (wind), and maximum significant swell and wave height (wave) are identified in the ERA5 reanalysis dataset (Hersbach et al., 2020) from January 1980 to December 2019. The data have a temporal frequency of 6 hours, with accumulations or maxima computed for each 6 h time interval centered around hours 00, 06, 12, 18 UTC, and are interpolated to a horizontal resolution of $0.5°$. Grid-point values exceeding the local 98[th] year-round percentile and a minimum threshold of 2 mm for rain, 10 ms$^{-1}$ for wind and 2 m for waves constitute our sample of uni-variate extremes (see maps of thresholds in Fig. SM1[2]); these are classified as rain–wind or wave–wind

---

[2]SM figure label refers to the Supplementary Material document





compound extremes when occurring at the same time step and grid point. Moreover, the results are not sensitive to the specific choice of percentile threshold (cf. Fig. 4 and Fig. SM3, using a 98[th] and 95[th] percentile threshold, respectively).

We adopt year-round thresholds for the identification of extremes to obtain consistent and potentially impactful events across seasons, with minimum thresholds intended for neglecting those occurrences which, despite belonging to the upper quantile, are very weak in intensity (e.g., rainfall over the Sahara, see Fig. SM1). Weather extremes from the upper two percentiles, although allowing for a limited detection of summer compounds of both types, provide an adequate sampling of compounds during the other seasons. The selection comprises extremes with varied impact potential: while the effects of extreme 6 h rain

depend on its accumulation over hourly to seasonal time spans (e.g., Guzzetti et al., 2008; Froidevaux et al., 2015; Kilsdonk et al., 2022), wind events of this type are damage-relevant (Klawa and Ulbrich, 2003). It is important to bear in mind that, although not all statistically extreme meteorological conditions have serious societal or environmental consequences, extremes that are *per se* moderate occasionally combine to produce severe impacts (Seneviratne et al., 2012).

For estimating rainfall we use ERA5 total precipitation, corresponding to the sum of the large-scale and convective precip-

itation generated by the ECMWF Integrated Forecasting System (IFS). Since the IFS grid (∼30 km spacing in the horizontal) does not resolve convective processes, these are parametrised; note that convection can be relevant for the (concomitant) occurrence of small-scale rain, wind and wave extremes. Lavers et al. (2022) visually compare ERA5 patterns with four observed impactful rainfall extremes and show a satisfactory performance of the reanalysis, notwithstanding a negative bias in representing the highest precipitation totals. The temporal correspondence between observed and ERA5 intense precipitation events is

estimated to be around 40-50% by Rivoire et al. (2021). Owen et al. (2021) analyse the geographical and temporal distribution of compound rain–wind extremes and report a high consistency between ERA5 and observed datasets. The simulation of wind gust and waves in ERA5 has improved with respect to ERA-Interim (Hersbach et al., 2020), although negative wind gust biases are measured inland and over orography because of the incorrect representation of wind channeling (see Minola et al. (2020); Obermann-Hellhund (2022) and ECMWF user guide [3]) and negative wave biases are reported, especially around coastal areas

(Fanti et al., 2023).

## 2.2 Mediterranean cyclones

We consider cyclones over the Mediterranean region in the period 1980–2019. The cyclone tracks are a result of the efforts of the MedCyclones COST Action [4] to produce a reference dataset of cyclone intensity and position (i.e., the value and location of the SLP minimum) within the Mediterranean region. The composite tracking approach described in Flaounas et al. (2023)

combines ten different cyclone detection and tracking methods applied to the ERA5 reanalysis (Hersbach et al., 2020) to identify cyclones spending at least 24 h in a broad Mediterranean domain (20°–50° N and 20° W–45° E). Although the tracks are computed on the original ERA5 temporal and spatial resolution, in this study, for consistency with the compound data described in Section 2.1, we interpolate the position onto a 0.5° horizontal grid and select the time steps coinciding with the hours 00, 06, 12, 18 UTC.

---

[3]https://confluence.ecmwf.int/display/CKB/Windstorm+footprints%3A+Product+User+Guide
[4]See website at https://medcyclones.eu/



As in recent articles using the Flaounas et al. (2023) cyclone dataset (see Givon et al., 2023; Rousseau-Rizzi et al., 2023), we select the tracks with confidence-level of 5, i.e., those satisfying an agreement among a minimum of five detection methods for at least 12 h in the cyclone's lifetime. The selection yields 3190 cyclones from 1979 to 2020, excluding many shallow heat lows occurring in summer or in the transition seasons (Givon et al., 2023).

### 2.2.1 Cyclone clusters

A clustering of the aforementioned Mediterranean cyclones based on their upper-level PV characteristics is performed by Givon et al. (2023). They obtain nine cyclone classes (identified by numbers), which differ by construction in terms of upper-level dynamics and synoptic-scale drivers, but also reveal a dominant seasonality and spatial distribution, together with distinctive tropospheric characteristics and surface impacts. Clusters 1, 2 and 4 are winter baroclinic cyclones (stage-A and -B lee lows and cyclones developing from Rossby wave breaking, respectively), clusters 5 and 8, peaking in the transition seasons, correspond to anticyclonic and cyclonic wave breaking, while clusters 3 and 7, showing maximum occurrence in spring, are associated with long wave cut-off lows and daughter lows, respectively; clusters 6 and 9, often land based, are shallow summer lows (Sharav heat low and short-wave cut-off low, respectively). For a detailed description of the cyclone classes and for their relevance in terms of weather compounds we refer to Givon et al. (2023) and Rousseau-Rizzi et al. (2023).

### 2.3 Dynamical features

Mediterranean cyclones, as extratropical cyclones in general, are circulation anomalies which displace air masses and create strong temperature gradients (Flaounas et al., 2022; Carlson, 1980). In this work we consider three types of air streams and fronts typical of cyclonic systems, namely dry intrusions (DIs), warm conveyor belts (WCBs) and cold fronts (CFs), and we analyse their links to compound extremes. A clear schematic of their organisation around Mediterranean cyclones, and of the related surface impacts, is provided in Fig. 17 of Raveh-Rubin and Wernli (2016).

We use existing datasets of objectively identified dynamical features to label feature masks, as in the following:

- non-zero WCB trajectory density. Lagrangian trajectories are identified in the ERA5 reanalysis based on an ascent of at least 600 hPa within 48 hours and on the presence of a nearby extratropical cyclone (Madonna et al., 2014; Heitmann et al., 2023; Wernli and Davies, 1997; Sprenger et al., 2017); the areas of low-level inflow (up to 800 hPa) and mid-level ascent (up to 400 hPa) are taken into account. Because Heitmann et al. (2023)'s global cyclone tracks include a different set of systems within and outside of the Mediterranean region, and because they use a separate WCB–cyclone attribution method, Mediterranean WCBs are not strictly associated with a Mediterranean cyclone track;

- non-zero trajectory density of DI outflow[5]. DI trajectories, identified in ERA5 based on a descent of at least 400 hPa within 48 h (Raveh-Rubin, 2017), are selected in the lower troposphere (up to 700 hPa), as in Catto and Raveh-Rubin (2019);

---

[5]The original DI dataset is smoothed using 2d convolution on a 4-by-4 symmetric kernel, in order to connect fragmented masks into single DI objects.





– CF lines, detected in ERA5 reanalysis following Hewson (1998) and Sansom and Catto (2022), are extended by a 2.5°
        distance to obtain a 2-dimensional object around each front as in Catto and Pfahl (2013).

Feature masks of WCBs (inflow and ascent), DI outflows and CFs correspond to connected objects of unit-valued grid points[6]
satisfying the conditions above; examples are shown in the left panel of Fig. 1. Furthermore, based on the "impact area criteria"
described in Section 3, each WCB, DI or CF mask is attributed - or not - to any of the co-occurring Mediterranean cyclones;
cyclone-less features are labeled as "stray".

## 2.4 Notation

We express the absolute frequency of an event $e$ as $p(e) = n(e)/N$, where $N$ is the total number of time steps in the period
considered. The frequency of $e$ conditional on the occurrence of another event $f$ is represented by $p(e\,|f) = n(e \wedge f)/n(f)$,
where $\wedge$ indicates co-occurrence (spatial and temporal overlap) of two events, $\vee$ indicates occurrence of at least one of two
events.

## 3 An *impact area* for Mediterranean cyclones

The *impact area* is the region around a cyclone center within which an event (e.g., a compound extreme) is associated to the
presence of that cyclone. In the literature on extratropical cyclones many possibilities are explored:

        – geometric definitions using a specific radius around the cyclone centre (e.g., a 1000 km radius in Owen et al. (2021), a
165        1100 km radius in Utsumi et al. (2016), or variable radii in Hepworth et al. (2022));

        – dynamical definitions based on closed SLP contours, as in Pfahl and Wernli (2012); Papritz et al. (2014); Dowdy and
          Catto (2017);

        – in some cases, the contribution of dynamical features related to the presence of the cyclones are also taken into account
          (e.g., cold fronts in  Papritz et al., 2014), or advocated for (Pfahl and Wernli, 2012).

When defining a Mediterranean cyclone's impact area, the large variety of topographic settings must be taken into account.
In fact, although Mediterranean cyclones are on average weaker, smaller and shorter-lived than their North-Atlantic equivalents
(Trigo et al., 1999; Trigo, 2006; Čampa and Wernli, 2012; Campins et al., 2011), the interaction of the induced atmospheric flow
with coastal boundaries and orographic features affects the spatial distribution of the surface impacts (Pfahl, 2014; Houze Jr,
2012; Obermann-Hellhund, 2022; Owen et al., 2021; Flaounas et al., 2019), as compared with a conceptual airflow model
of an extratropical cyclone over the ocean (Carlson, 1980; Wernli and Davies, 1997; Schultz, 2001). As discussed in Jansa
et al. (2001) and Pfahl (2014), the relative position the cyclone centre with respect to the warm Mediterranean sea, heating and
moistening the atmosphere, and to the coast plays a fundamental role in the understanding of the character of the impacts.

---

[6]A region of unit-valued grid points connected with each other via a structuring element [[111][111][111]], allowing for diagonal connections.




Reconciling the above aspects, and based on the observation of the synoptic maps and the extreme impacts of the eighty-six Mediterranean cyclone tracks occurring in year, we define an impact area for Mediterranean cyclones (labeled IA01) as a *central area*, a circle of 1000 km radius from the cyclone centre[7], eventually extended by the feature masks satisfying the conditions below:

1. *cold front* masks intercepting a circle of 500 km radius from the cyclone centre;

2. *warm conveyor belt* masks intercepting a circle of 500 km radius from the cyclone centre;

3. *dry intrusion* masks intercepting a circle of 1000 km radius from the cyclone centre.

The interception criteria are satisfied when the concerned feature masks overlap in at least one grid point (see an example of IA01 for cyclone Rolf in Fig. 1).

We note that changing the reference area to a central circle of 500 km radius extended by the same dynamical features listed above (labeled IA02) does not determine substantial qualitative changes in the results (cf. Figs 4, 7 and Figs A1, A2). However, the differences in the frequency statistics obtained with the two definitions, discussed in Appendix A, show how the wider impact area (IA01) is well suited for detecting cyclone-related wave–wind compounds (c.f. Raveh-Rubin and Wernli, 2015, 2016), while the more stringent definition (IA02) performs well in identifying cyclone-related rain–wind compounds. The suitability of IA02 for rain impacts partially verifies the approach by Flaounas et al. (2018), who attributed connected masks of non-zero precipitation to Mediterranean cyclones when the former intercepted a small (∼250 km) radius around a storm centre.

In this work we use the larger impact area definition (IA01) for three reasons: (i) it allows the attribution of extreme winds and waves occurring at a considerable distance from the cyclone centre (Nissen et al., 2010; Pfahl, 2014; Field and Wood, 2007), (ii), in spite of being loose for the identification of cyclone-related precipitation extremes, it captures distant rainfall events resulting from the interaction of the cyclonic flow with orography (Pfahl, 2014; Owen et al., 2021), and (iii) for comparability between rain–wind and wave–wind cyclone statistics. For the sake of future cyclone attribution studies, we mention that the results obtained using IA01 largely correspond to those using a fixed 1000 km radius impact area.

## 4  Frequencies of compound extremes and Mediterranean cyclones

The frequency of compound extremes is unevenly distributed across regions and seasons. Concomitant rain–wind extremes are most frequent in winter (Fig. 2), as a result of a winter peak in the uni-variate extremes (Fig. SM2) and in the compounding ratio ($p(R\wedge W)/p(R\vee W)$, see Fig. SM4). The winter R∧W events occur most often in the eastern Mediterranean, specifically along the coastlines of Israel and North-Africa and the eastern Adriatic sea. In agreement with Raveh-Rubin and Wernli (2015) (their Figs 3 and 4), the autumn spatial maximum in R∧W frequency is shifted to the central and western Mediterranean, where differences from the winter season are small. The spring frequency, limited by the number of extreme rainfall events

[7]A fixed-radius area is here preferred to a SLP-based dynamical area because of Mediterranean orography inducing uncertainties in the SLP field (Flaounas et al., 2015).



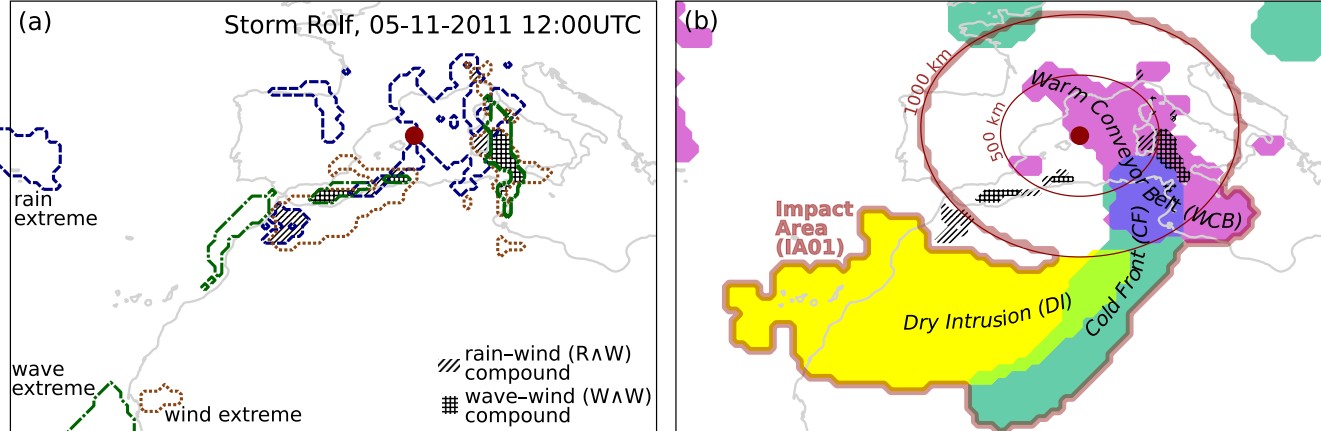

**Figure 1.** Cyclone Rolf's centre at 12 UTC on 5 November 2011, indicated by a dark-red dot, (a) with extreme events and compound events, and (b) with cyclone features and compound events. Rain, wind and wave extremes are denoted with blue dashed, brown dotted and green dash-dotted contours; rain–wind compounds by slanted-line hatching, wave–wind compounds by square-grid hatching. Dry intrusions, cold fronts and warm conveyor belts are shown as yellow, aquamarine and purple objects, while the semi-transparent red contour encircles the cyclone's impact area IA01 (definition in Section 3)

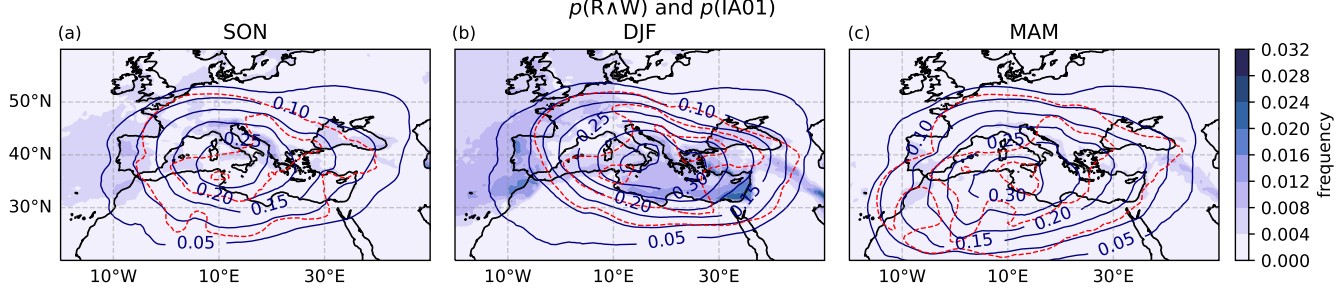

**Figure 2.** Frequency of R∧W compound ($p$(R∧W), shading) and of cyclone impact area ($p$(IA01), blue contours) in (a) autumn - SON, (b) winter - DJF and (c) spring - MAM. The frequencies correspond to the number of occurrences divided by the number of time steps within each season. The difference between $p$(IA02) and $p$(IA01) is represented by negative dashed red contours at 0.05 interval

(Fig. SM2(c)), is substantially lower everywhere (on average[8] about a half of the autumn and a quarter of the winter frequencies). The R∧W compounding ratio is generally highest over coastal regions facing north-west (Fig. SM4) and over orography
210 (consistent with Fig. 2b in Owen et al. (2021) and Fig. 4a in Martius et al. (2016)).

Concomitant wave and wind extremes also occur must frequently in winter (Fig. 2), but, differently from R∧W compounds, show a greater spatial homogeneity within the basin (Fig. 3). There is a shift in the position of the maximum frequency from the western Mediterranean in autumn to the eastern Mediterranean and the Black Sea in winter. Spring W∧W extremes display

---

[8]Average over the Mediterranean region within 28°–50° N, 10° W–45° E, excluding white masked areas.



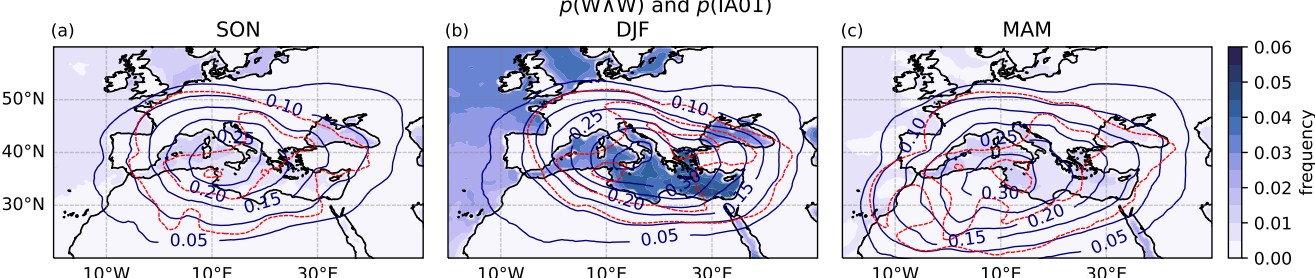

**Figure 3.** As in Fig. 2 for W∧W compound (note difference in colour scale)

relatively small spatial variability with an average incidence similar to autumn values. The high compounding ratio across seasons (Fig. SM5) is a consequence of the strong physical dependence of wave height on the surface wind field (e.g., Komen et al., 1996). Nevertheless, both the absolute frequency and the compounding ratio of W∧W events are generally higher in the Mediterranean than in the western Atlantic (cf. Fig. 1 in Catto and Dowdy, 2021).

The winter peak in compound extremes coincides with the highest frequency of cyclones ($p$(IA01), blue contours in Fig. 2). The Mediterranean cyclone density maximum shows a peak over the Italian peninsula in the autumn season, it extends towards the south east in winter and towards the south west (extending to northwestern Africa). The cyclone density distribution is spatially smoother and displays different regional and seasonal patterns compared to the compound frequency; other factors, beyond cyclone frequency, account for the distributions of weather compounds.

## 4.1 Rain–wind compounds (R∧W)

The frequency of R∧W events associated with a cyclone (Fig. 4(a)-(c)) is highest in winter, with a 1 to 10% frequency over most of the basin peaking over the eastern Mediterranean and the northern coast of the African continent, and lowest in spring - generally below 1%, when cyclones are known to produce weaker rainfall amounts (Flaounas et al., 2019).

The presence of a nearby cyclone increases the compound frequency (Fig. 4(d)-(f)), particularly in the transition seasons compared to winter and in the eastern and central Mediterranean, where a 10-fold amplification is not uncommon. The maps in Fig. 4(g)-(i), representing the proportion of cyclone-related R∧W compounds, show that autumn and spring events frequently occur with a cyclone (on average[9] 50% and 58% of the R∧W events co-occur with cyclones in autumn and spring, compared to 48% in winter; differences are locally larger). Note that the relation between the quantities $p$(IA01| R∧W) and $p$(R∧W| IA01) in Fig. 4 follows Bayes' theorem (i.e., $p$(IA01| R∧W)$\cdot p$(R∧W)$= p$(R∧W| IA01)$\cdot p$(IA01)), which states that the fraction of cyclone-related compounds increases with the ratio of cyclones over compounds. Hence, in winter the distinct peak of $p$(R∧W) compared to $p$(IA01) implies a smaller proportion of cyclone-related events. The occurrence of winter R∧W events outside of cyclone impact areas, or overlapping with stray[10] cold fronts and stray warm conveyor belts, is particularly high in the eastern

---

[9]See footnote 8.

[10]The word "stray" is used to indicate features that do not fulfill the criteria for joining a Mediterranean cyclone's impact area, as described in Section 3.



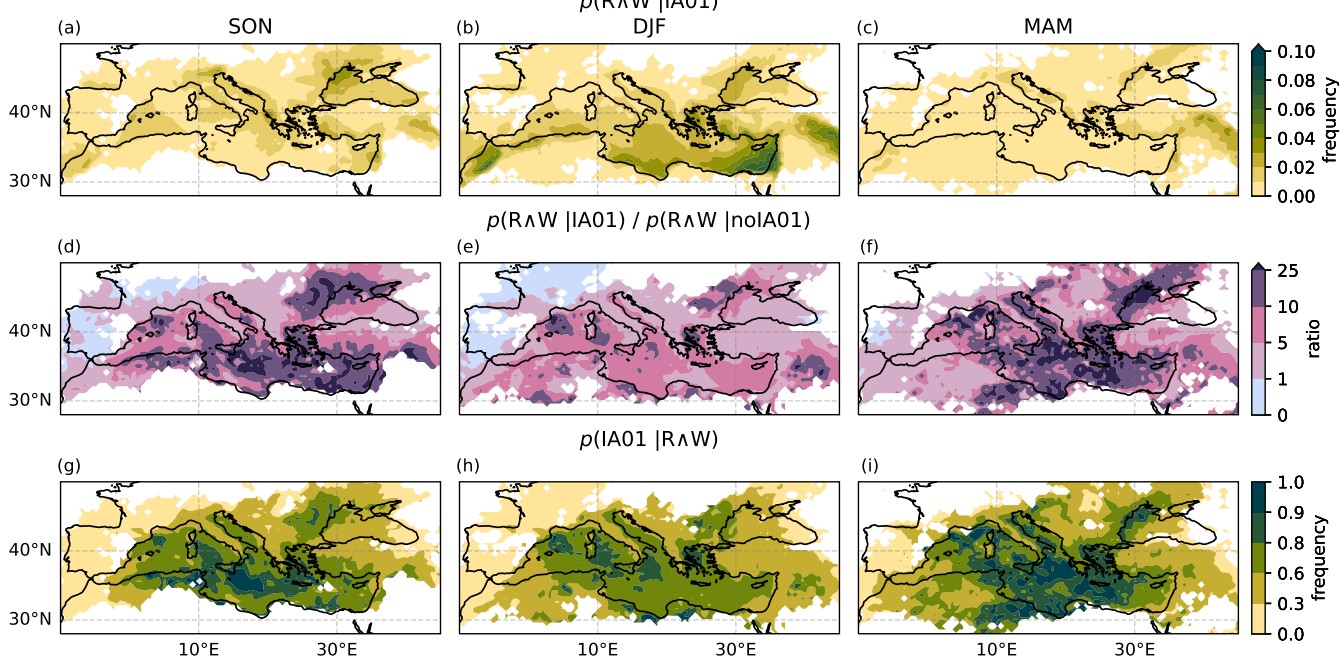

**Figure 4.** (a-c) The frequency of R∧W compound conditional to the presence of a cyclone, (d-f) the ratio of R∧W compound frequency during cyclone occurrence over the R∧W compound frequency during times when cyclones do not occur, (g-i) the cyclone frequency conditional to the presence of R∧W compounds. Seasonal results for autumn - SON, winter - DJF and spring - MAM are displayed on the left, centre and right panels, respectively. Grid points displaying less than four (R∧W |IA01) events are masked in white

and southern sectors of the Mediterranean (in Fig. 5 hatching-free areas show where at least 10% of compounds overlap with stray dynamical features).

We separate the influence of the Mediterranean cyclones' dynamical features, as in cold fronts (CF-IA), regions of warm conveyor belt inflow (WCBin-IA) and ascent (WCBas-IA), and dry intrusions (DI-IA), from the rest of the impact area (IA-noDF). Figure 6(a) shows the feature co-occurring most frequently with R∧W events, Figure 6(b) represents the feature maximising the R∧W event incidence; both are computed around the year because seasonal differences are small. Over the whole Mediterranean region, the occurrence of R∧W events is maximised below the ascending branch of WCBs (Fig. 6(b)), while WCB inflow is more important over land and mountain reliefs during the winter season (not shown). On the other hand, when selecting the most frequent feature during R∧W compounds (Fig. 6(a)), WCB ascent is still predominant in the northern Mediterranean close to mountain reliefs or on sea-based grid points facing west- or south-oriented coasts (see also Fig. 7c for precipitation extremes in Pfahl et al., 2014). CFs shows highest frequency on the rest of the basin (see also Fig. 4b for precipitation extremes in Catto and Pfahl, 2013), except around the North-African coasts, where the impact area without dynamical features is relevant. The differences between panels (a) and (b) of Fig. 6 derive from the fact that in (b) the results are normalised by the features' absolute frequencies (see Fig. SM6).



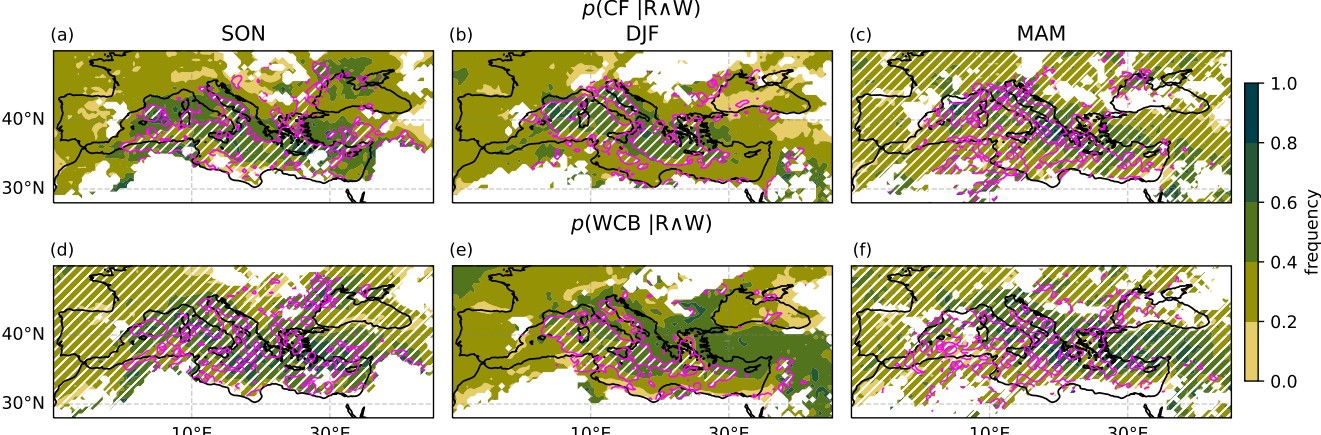

**Figure 5.** Frequency of (a-c) CFs and (d-f) WCBs conditional to R∧W compound occurrence in (a,d) autumn - SON, (b,e) winter - DJF and (c,f) spring - MAM. Note that all CFs and WCBs, independently of their being part of a cyclone's impact area, are considered; magenta contours and slanted white hatching identify regions where more than 90% are included in a cyclone's impact area (definition in Section 3). Note that, since compounds occurring within superposing CFs and WCBs contribute to the statistics of both, the sum of their conditional frequencies can exceed 100%

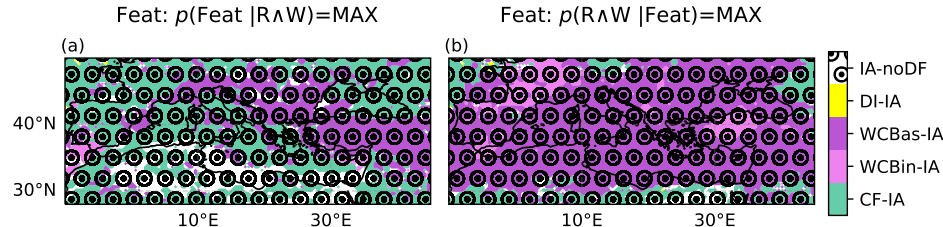

**Figure 6.** (a) The most frequent cyclone feature during R∧W compounds; (b) the cyclone feature associated with highest occurrence of R∧W compounds. We compute the conditional probability maxima amongst five cyclone features: the cold front (CF-IA), the warm conveyor belt inflow (WCBin-IA) and ascent (WCBas-IA) regions, the dry intrusion (DI-IA), and the 1000 km radius around the cyclone centre after removal of the aforementioned dynamical features' masks (IA-noDF). Compounds occurring within superposing CFs, WCBs and DIs contribute to the statistics of all the objects concerned. Because of a weak variability across seasons, the results are provided using data over all months

## 4.2 Wave–wind compounds (W∧W)

250

In the cyclone neighbourhood, wave–wind extremes co-occur much more frequently than rain–wind extremes (cf. Fig. 7(a)-(c) and Fig. 4(a)-(c)). The main similarity between the two types of compounds is the higher compound frequency for winter cyclones (in Fig. 7(b) W∧W reach and locally surpass a 10% frequency), and the tendency for the spatial peak in the cyclone





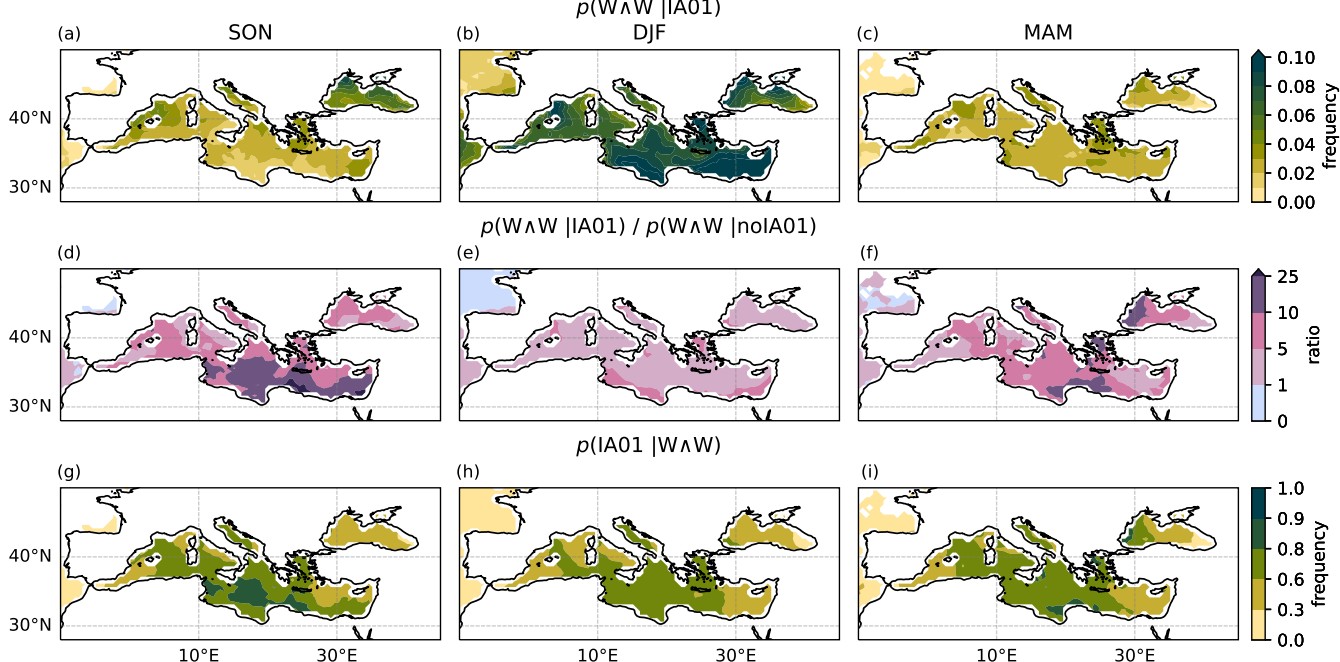

**Figure 7.** As in Fig. 4 for W∧W compound

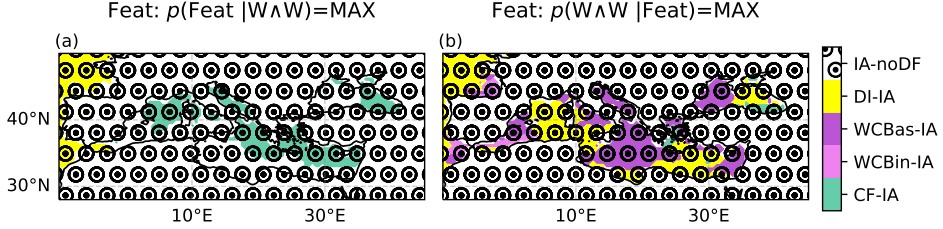

**Figure 8.** As in Fig. 6 for W∧W compound

compound density to shift from the western Mediterranean in autumn to the south-eastern basin in winter. This second aspect
is coherent with the geographical shift in W∧W and impact area frequency (Fig. 3).

The proportion of W∧W events associated with cyclones peaks in the central part of the basin and is quite constant across
seasons (on average[11] ∼50%, Fig. 7(g)-(i)). Moreover, the values of W∧W co-occurrence with cyclones, rarely exceeding an
80% level, are smaller compared to those for R∧W events, when the percentage locally surpasses the 90% level (cf. Fig. 7(g)-
(i) and 4(g)-(i)). This suggests cyclones to be less relevant for W∧W than R∧W for compounding. The W∧W compound
frequency increases approximately by a factor 5 in the cyclone neighbourhood (Fig. 7(d)-(f)) and exhibits a strong inter-
seasonal variability that is likely related to a larger fraction of winter events occurring with stray dynamical features (Fig. SM8).

[11]See footnote 8.



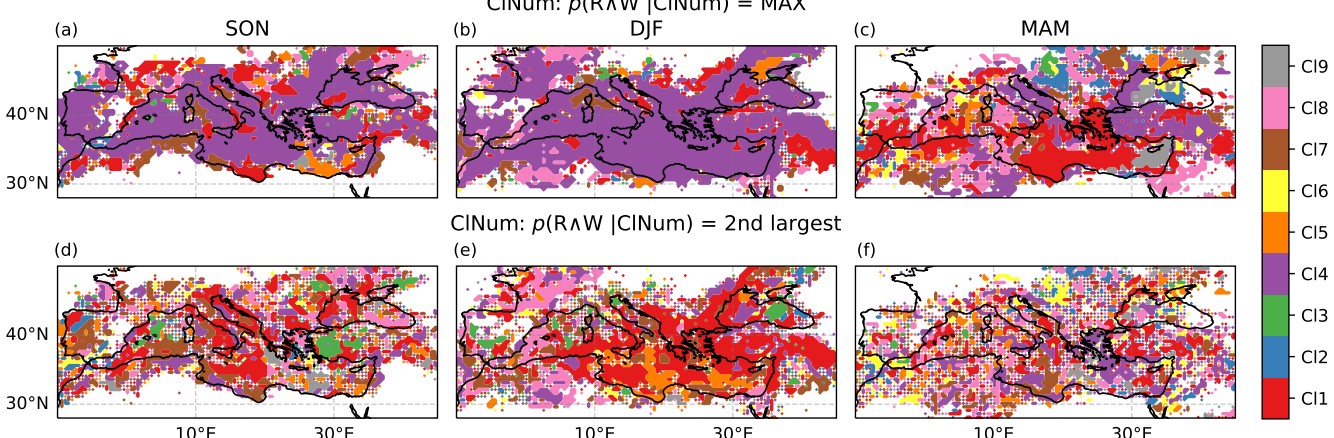

**Figure 9.** The first and second cyclone cluster that maximise the R∧W compound frequency in (a,d) autumn - SON, (b,e) winter - DJF and (c,f) spring - MAM. Note that compounds occurring within multiple cyclones' IA01 contribute to the frequencies of all the relevant clusters. Grid points displaying less than four (R∧W |IA01) events are masked in white; note that this results in weak statistics presenting irregular and intermittent colour patches

Dry intrusions, which climatologically drive intense surface wind gusts (Raveh-Rubin, 2017) and induce impacts on surface and atmospheric conditions (e.g., dust uplift, see Fluck and Raveh-Rubin, 2023), are the most important dynamical feature for the occurrence of wave–wind compounding over the western and eastern parts of the basin (Fig. 8(b)). WCB ascent is important

265 for W∧W occurrence in the central Mediterranean (comprising the Adriatic Sea and most of the Tirrenean and Ligurian Seas), south-east of Spain and in the eastern-most part of the basin. In contrast, CFs are the most frequent feature to be detected during W∧W events in the northern Mediterranean, replaced by the feature-free 1000 km radius in the southern Mediterranean (Fig. 8(a)). Both features show high absolute frequencies (Fig. SM6).

### 4.3 Cyclone clusters relevant for compounding

270 In this section we refer to the Mediterranean cyclone clusters described in Givon et al. (2023). While the association of each cluster with surface compounds from a cyclone centric view is discussed in Rousseau-Rizzi et al. (2023), in Figs 9 and 10 we show the two cyclone clusters with the highest frequency of R∧W and W∧W from a (geographical) Eularian perspective.

Baroclinic clusters 4 and 1 are, in order of importance, the most relevant for both types of compounds in autumn and winter. In winter we also find a strong regional association of cluster 8 with R∧W events over the western Mediterranean, and of

275 cluster 5, typical of transition seasons, with both compound extremes in the central and south-eastern Mediterranean (the R∧W region in Fig. 9(b),(e) is to the east of the W∧W region in Fig. 10(b),(e), as expected in the conceptual model of Mediterranean cyclones' impacts by Raveh-Rubin and Wernli (2016)). In autumn cluster 7, corresponding to the daughter-low type and often originating over North Africa in a relatively dry environment, is important in the south-western Mediterranean and in the eastern-most part of the basin, but only relative to W∧W occurrence (Fig. 10(a),(d)).



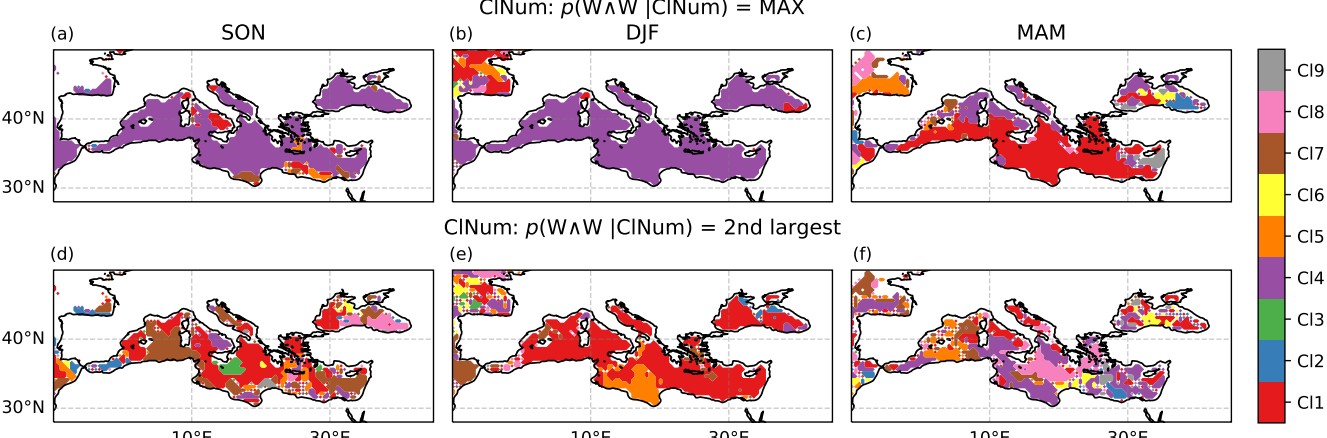

**Figure 10.** As in Fig. 9 for W∧W compound

In spring the dominant-cluster distribution is more diverse, with cluster 1 maximising the compound occurrence over most of the basin. Number 4, because of its frequency peaking earlier in winter, becomes the second most important cluster (first in the central-north Mediterranean). Cluster 9, corresponding to cut-off lows, acquires relevance for both types of compound in the eastern Mediterranean and in the Black Sea, while in the western Mediterranean the W∧W events are often linked to clusters 5 and 7, the R∧W events to the moister cluster 8. The sparse distribution of cluster 8 extends to a wide central Mediterranean region, where it is also associated with frequent W∧W events (Figs 9(c),(f) and 10(c),(f)). Cluster 2, typically comprising large, baroclinic winter cyclones, although identified by Rousseau-Rizzi et al. (2023) as the cluster with the highest simultaneity and overlap of rain and wind extremes, is associated with the highest R∧W compound frequency only in spring over the Balkan peninsula. This is likely a result of its relatively low occurrence frequency compared to clusters 1 and 4 (Fig. SM7).

We note that clusters 1 and 4, dominant across regions and seasons, show an average R∧W and W∧W frequency that is only slightly higher than the aggregated statistics in Figs 4(a)-(c), 7(a)-(c), whereas summer clusters 6 and 9 and cluster 2, maximising compound frequency over limited domains, show an amplification of the overall cyclone compound frequency by a factor generally larger than 5. The other clusters, typical of the transition seasons, show an intermediate increase in R∧W and W∧W frequency (factors close to 3) compared to the aggregated statistics. The comparisons are computed seasonally over the region where each cluster maximises the compound probability, i.e., areas filled with the corresponding colour in Figs 9(a)-(c) and 10(a)-(c).

## 5   Discussion

Frequency maps of rain–wind and wave–wind clustering show a pronounced winter peak and an eastward shift during the transition from autumn to spring (Fig. 2), following the density of the Mediterranean cyclones' impact area but with a more





 winter cyclones (Figs 4(a)-(c) and 7(a)-(c)), and only at a second order by changes in the number of cyclones.

During the transition seasons compounds - R∧W in particular - happen mostly within the impact area of a cyclone (Figs 4(g)-(i) and 7(g)-(i)), while in winter they have a weaker relation to cyclones and happen more frequently with stray dynamical features (cf. Figs 4 and 5, Figs 7 and SM8). This last result is consistent with Owen et al. (2021), who identify winter compounds in cold frontal regions far from storm centres; they also cite atmospheric rivers as a contributing factor for driving winter R∧W extremes (see also Hénin et al., 2021, for regional patterns). The high winter AR frequency in the region and the associated potential increase in the likelihood of R∧W events support such hypothesis (Guan and Waliser, 2015; Waliser and Guan, 2017); nevertheless, the association of atmospheric rivers (ARs) and rain–wind compounds around the Mediterranean basin and the superposition of ARs and warm conveyor belts (see discussion in Ralph et al., 2017) are yet to be explored systematically.

Geographical peculiarities, listed in the following, are more pronounced for R∧W compared to W∧W compounds, because of varying seasonal and regional moisture availability and of the strong effects of orography and coastlines on precipitation (e.g., Flaounas et al., 2019; Jansa et al., 2001; Pfahl, 2014). We specify that the statistical links between W∧W events and cyclones in the north-western Mediterranean basin (shown in Fig. 7) are liable to be sensitive to the inclusion of Atlantic cyclones, inducing strong wind impacts in the region (Pfahl, 2014; Nissen et al., 2010).

- The Gulf of Lion shows a year-round minimum in R∧W compounds-over-extremes ratio (Fig. SM4) and a maximum in the W∧W ratio (Fig. SM5). The rain–wind compounding ratio is related to dry northerly mistral-wind events, which bear no rainfall, yet are known to induce lee-cyclogenesis in the form of Genoa lows; these systems carry intense precipitation elsewhere, mainly over the Italian and Balkan peninsulas (Givon et al., 2021). The maximum in W∧W compounding ratio is related to the high wind intensity in the region (e.g., note the peak in the 98th percentile wind gust in Fig. SM1(e)) and to the physical dependence of the wave height on the wind field (e.g., Komen et al., 1996).

- In Eastern Spain, south-western France and in the eastern alpine region a low proportion of R∧W events is associated (year-round) with Mediterranean cyclones (Fig. 4(g)-(i)). Despite the exclusion of Atlantic tracks, our results correspond qualitatively to those considering global storm track by Owen et al. (2021). They relate the low co-occurrence of cyclones and R∧W compounds north of the Mediterranean (their Figs 10, 11, 12) to the fact that regional precipitation extremes are usually brought by Mediterranean cyclones, while wind extremes are often attributed to Atlantic weather systems. Similar findings are discussed in Pfahl (2014) and Nissen et al. (2010).

- The relation between cyclones and compounds is strong in the north west of the Black Sea, notwithstanding the distance from the Mediterranean (Fig. 4). The reason for this lies in a secondary peak in Mediterranean-cyclone track density over the former basin (see Fig. 13(b) in Flaounas et al. (2023)).

Regional differences in compound–cyclone statistics are further influenced by the varying size and type of the Mediterranean cyclones. For example, the high compounding frequency around winter cyclones in the south eastern Mediterranean (Figs 4 and 7) can be attributed to a favourable geographical setting - the coast is exposed to the north west, the direction from which



most weather systems propagate (see also Fig. 4a in Martius et al., 2016), but also to a small cyclone size favouring the superposition of wind and rain/wave impacts close to the core of the system (following Fig. 10 and the related discussion in Raveh-Rubin and Wernli (2015)). Our results confirm Raveh-Rubin and Wernli (2015)'s view by indicating a high ratio of
cyclone–compound co-occurrence in the presence of a fixed 500 km-radius area around the cyclone (not shown).

The clustering of Mediterranean cyclone introduced in Givon et al. (2023) allows us to study the spatial distribution of the cyclone types associated with highest compound frequency.

  – Clusters 1 and 4 , typical of strong baroclinic cyclones, are those with the strongest connection with regional rain–wind (and wave–wind) compounds across seasons, probably because of the high rainfall rates and of the wide spatial extension
of their surface impacts (see Figs 10 and 11 in Givon et al. (2023) and Fig. 9 in Rousseau-Rizzi et al. (2023)). The winter peak in their seasonal distribution helps to explain the maximum weather compounding around winter cyclones (Fig. 4).

  – Cluster 8 cyclones, relatively stationary and characterised by strong convective activity (Givon et al., 2023), despite being associated with surface hazards of limited spatial extension (see Figs 10 and 11 in Givon et al. (2023)), shows a high likelihood of R∧W compounds across multiple seasons and regions.

– Cluster 5 and 7 cyclones result from the interaction of PV streamers with Mediterranean and North-African topography.Cluster 7 is relevant for W∧W extremes, cluster 5 for both compounds in winter, for W∧W in spring.

  – Cluster 9, made up of cut-off lows typical of the extended summer season, emerges in spring over the eastern Mediterranean and is connected with both types of compounds.

The results of the cluster analysis are independent of the specific definition of impact area (Section 3 and Appendix A).
Additionally, our study identifies variations in the cyclone-related dynamical features associated with the two types of compound extremes. The presence of WCB ascent maximises the likelihood of R∧W events across the Mediterranean region, although a frequency-weighed analysis suggests the importance of diverse air streams in the central and northern Mediterranean (CFs over the sea, WCBs over land and close to west- and south-facing coastal boundaries), of the dynamical-feature-free impact area (and of the cyclone center, not shown) along the southern Mediterranean coast (Fig. 6). The results for the south-
ern coast are likely linked to a smaller size of the cyclones (Raveh-Rubin and Wernli, 2015), and, as discussed in Ziv et al. (2010), to the presence of dry, weakly-precipitating WCBs from North Africa. W∧W events are most probable with DIs and WCBs, although they co-occur frequently with CFs in the northern Mediterranean, and with the dynamical-feature-free impact area in the southern Mediterranean (Fig. 8). Results remain qualitatively unchanged when considering all dynamical features independently of their inclusion in a cyclone's impact area.
The high superposition of compounds and WCBs is in contrast with the exclusive rainfall impacts attributed to such air stream (Madonna et al., 2014). However, the vertical overlap of the WCB with cold frontal zones (Catto et al., 2015), often characterised by frontal convection (and wind gusts) or by the presence of a pre-frontal low-level jet (see Fig. 5 in Wernli (1997), Figs 7 and 9 in Eisenstein et al. (2023)), or with the convective activity near the cyclone center (Flaounas et al., 2015), likely corresponds to where many of the R∧W and W∧W extremes happen over the sea (e.g., Fig. 10(a) in Raveh-Rubin





and Wernli (2015) and Fig. 9 in Rousseau-Rizzi et al. (2023)). Moreover, coastal and orographic wind channelling and uplift (Carrera et al., 2009) combined with higher precipitation efficiency over land and/or orography (Flaounas et al., 2019; Pfahl, 2014; Houze Jr, 2012) may explain the high overlap of R∧W compounding and WCBs next to topography (cf. compound pattern and brown isohypse in Fig. 9 of Rousseau-Rizzi et al. (2023)).

## 6   Conclusions

We quantify the relation between compound weather extremes (rain–wind and wave–wind) and the presence of nearby cyclones in the extended Mediterranean region. Based on a new definition of "cyclone impact area", i.e., a fixed radius around the cyclone centre and overlapping fronts and cyclone air streams (Section 3), we answer the research questions listed in the Introduction.

1. We verify that the presence of a cyclone increases the odds of compound incidence, particularly in the transition seasons - by a factor 5 at least - and for R∧W events. Winter cyclones, even if less determinant for the occurrence of compound extremes, display the strongest compound frequency (1-10% for R∧W, more than 5% for W∧W); differences between autumn and spring R∧W statistics are related to a lower moisture availability in the spring season. (Section 4)

2. R∧W and W∧W compounding tend to overlap with different dynamical features around a cyclone, showing a distinct regional dependence. In general, the probability of rain–wind extremes around cyclones is maximised in areas of warm conveyor belt ascent (∼7% of the cases), that of wave–wind extremes below dry intrusion outflow and warm conveyor belts (more than 10% of the cases). A frequency-weighed analysis also highlights the association between compounds and cold fronts in the north of the Mediterranean region, between compounds and feature-free cyclones in the south. (Sections 4.1, 4.2)

3. Baroclinic cyclones (clusters 1 and 4 from Givon et al. (2023)) maximise the occurrence of compounds across all seasons, and their peak occurrence in winter explains the highest cyclone compound density for the winter season (see point 1.). Other cyclone categories are relevant at a local and seasonal scale. For example, compact, stationary and convective cyclones - classified as cluster 8 - are associated with frequent R∧W extremes over different seasons and parts of the basin. (Section 4.3)

Crucial questions remain regarding the role of orography, coastal boundaries, wind channelling and convection in the generation of compound R∧W and W∧W events in the Mediterranean region. In particular, our results point to the importance
of warm conveyor belts (WCBs) for rain–wind and wave–wind compounding in the northern Mediterranean, even though WCBs are only known to induce abundant rainfall. Interactions of the WCBs with orography and coastal boundaries, or WCB superposition with frontal convection, pre-frontal low-level jets and cyclone centers, may be important for the generation of wind extremes. Additionally, the role of atmospheric rivers, to be considered together with potentially overlapping WCBs, in Mediterranean rain and rain–wind extremes deserves targeted analyses.

Future research could also separate compounds in events induced by the synoptic circulation or by deep convection along the frontal boundaries of the cyclone. Based on previous literature (e.g., Flaounas et al., 2018; Givon et al., 2023), we expect a





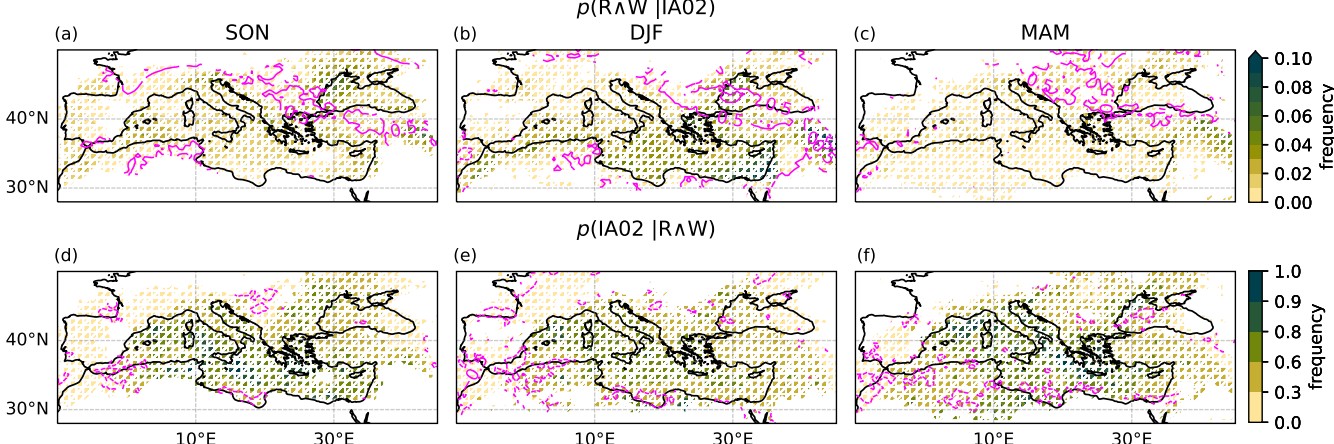

**Figure A1.** (a-c) The frequency of R∧W compound conditional to the presence of a cyclone, (d-f) the cyclone frequency conditional to the presence of R∧W compounds, using an impact area with a reference radius of 500 km (IA02). A difference between IA02 and IA01 frequencies by more than 25% of the IA01 value is denoted by white slanted-line hatching; a 50% difference is highlighted by white square-grid hatching within magenta contours (dashed for negative differences). Seasonal results for autumn - SON, winter - DJF and spring - MAM are displayed on the left, centre and right panels, respectively. Grid points displaying less than four (R∧W |IA02) events are masked out

different spatial extent and distribution (geographical and relative to the cyclone centre) of the two types of events. Additional insight on convective environments and hazards around Mediterranean cyclones (in a cyclone-centred perspective, following Givon et al. (2023) and Rousseau-Rizzi et al. (2023)) will be addressed in a separate paper.

Finally, we expect Mediterrenean cyclone-related weather compounds in a future warmer climate to change in distribution and intensity, together with changes in the cyclones' frequency and nature. Recent trends show a decrease in the number of strong baroclinic systems and an increase in the number of small-scale cyclones typical of the extended summer season (Givon et al., 2023), while end-of-XXI-century climate projections indicate a general reduction in the frequency and overall intensity of Mediterranean cyclones (Reale et al., 2022). The connection with changes in the number of weather compounds is yet to be
examined.

## Appendix A: Discussion on the choice of impact area

Various possibilities of impact areas were tested before establishing the definition of IA01 described in Section 3. Particularly interesting is the comparison between the results using IA01 and IA02, where the latter corresponds to a smaller reference area (radius of 500 km instead of 1000 km) extended by the same dynamical features as IA01 (see Section 3). The frequency
difference "$p(\text{IA02}) - p(\text{IA01})$" is shown in Fig. 2. Limit cases of IA01 and IA02 compound density correspond to:

  i. all cyclone compounds ($C$) captured by IA01 fall within IA02
     $p(C\,|\text{IA01-IA02}) = 0 \quad \Rightarrow \quad p(C\,|\text{IA02}) \gg p(C\,|\text{IA01})$ and $p(\text{IA02}\,|C) = p(\text{IA01}\,|C)$;



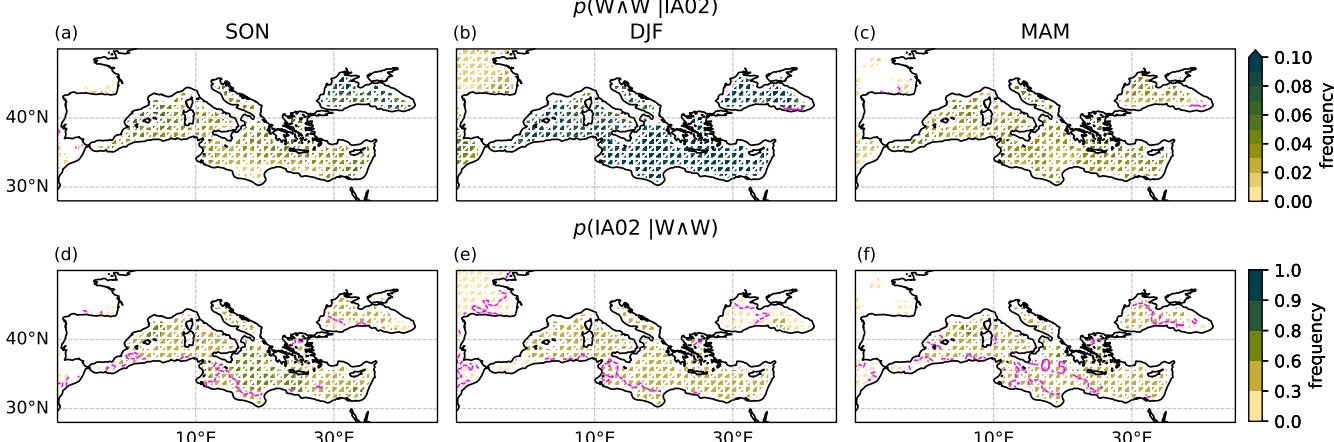

**Figure A2.** As in Fig. A1 for W∧W compound

    ii. the density of cyclone compounds is uniformly distributed within IA01

$$p(C\,|IA01) = const \quad \Rightarrow \quad p(C\,|IA02) = p(C\,|IA01) \text{ and } p(IA02\,|C) \ll p(IA01\,|C).$$

In (i.) the best choice is IA02, since IA01 overestimates the extension of the expected impacts and underestimates $p(C\,|IA01)$, the compound occurrence around the cyclone; in (ii.) IA01 is most adequate, since IA02 underestimates the area of influence and $p(IA02\,|C)$, the ratio of cyclone-related compounds.

    In Fig. A1 we show in conditional probabilities between cyclones and R∧W compounds obtained using IA02, while hatching indicates percentage levels of difference from the same quantities computed using IA01 (full IA01 statistics in Fig. 4). Differences between $p(R\wedge W\,|IA02)$ and $p(R\wedge W\,|IA01)$, positive and often exceeding the IA01 values by more than 25% (slanted-line hatching, panels (a)-(c) in Fig. A1), indicate a higher frequency of compounds within the smaller impact area. The largest differences in R∧W incidence are detected in the transition seasons, when cyclones are smaller and severe hazards are distributed closer to the cyclone centre (within IA02) (Givon et al., 2023), and in the region separating the Mediterranean and the Black Sea. Hatching along the North-African coast in Fig. A1(d)-(f) also highlights a substantial regional decrease in $p(IA02\,|R\wedge W)$ relative to $p(IA01\,|R\wedge W)$. This probably relates to IA02 missing distant compound events which result from the interaction between the large-scale cyclonic flow and the African topography.

    The comparison of IA01 and IA02 for W∧W events diverges substantially from that of R∧W compounds (cf. Figs A2 and A1). The probability of W∧W within IA02 (Fig. A1(a)-(c)) does not increase nearly as much as in the case of R∧W, while the choice of IA02 strongly reduces the proportion of cyclone-related compounds throughout the Southern Mediterranean and over the Black Sea (Fig. A1(d)-(f)).

    In summary, rain–wind compounds, showing a large increase in compound frequency close to cyclones when using IA02, are reminiscent of limit-case (i.) with a preference for IA02 (except over the Southern Mediterranean coast). Differently, wave–wind compounds, because of small differences in the compound frequency within IA02 and IA01, and of an important fraction



of compounds occurring outside IA02 but within IA01, are relatable to limit-case (ii.) and privilege IA01. This is consis-
tent with the expected geographical distribution of cyclone impacts, with intense rainfall and convective activity happening
close to the centre, strong wind and waves occurring further away. The result validates previous hazard-dependent choices of
Mediterranean-cyclone impact area (e.g., Flaounas et al., 2018); in this work, for the sake of methodological consistency across
results, we use the IA01 definition.

*Author contributions.*    AP, OM, SRR and JLC conceived the study. YG provided data and advise for the cyclone-cluster analysis. AP per-
formed the analyses and prepared the manuscript with contributions from all co-authors.

*Competing interests.*    One of the authors is co-editor for Weather and Climate Dynamics.

*Acknowledgements.*    AP is grateful for insightful discussions with Prof David Stephenson and Dr Matthew Priestley. The authors are thankful
to all the collaborators who produced and provided the CF and DI datasets, and to Prof Heini Wernli and Prof Micheal Sprenger for kindly
providing the WCB dataset. The research is funded by the Swiss National Science Foundation (SNF) Grant Number IZCOZ0_205461, and
contributes to the efforts of COST Action CA19109 "MedCyclones: European network for Mediterranean Cyclones in weather and climate".
SRR acknowledges funding from the Israel Science Foundation (grant number 1242/23) and the De Botton Center for Marine Science at the
Weizmann Institute.



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
