# Peer review of "Linking compound weather extremes to Mediterranean cyclones, fronts and air streams"

_EGUsphere, 2024_

## Referee Comment (RC1)

**Review Portal et al. WCD**

The paper by Portal et al. addresses an important thematic that in general deserves more attention by the community, which is the identification of dynamical drivers of compound extremes. In the paper, the authors focused on Mediterranean cyclones and compound rain-wind, wave-wind extremes, within the reanalysis period 1980-2019. They introduce a new definition of cyclone's impact area and assess how three different dynamical features contribute to the occurrence of the extremes. Then, they also quantify the most frequent cyclone type linked to the extremes and provide a discussion of their results.

The paper is sound, well written and therefore suitable for publication in WCD. Below you find some major and minor comments that I hope the authors will address in the review process.

Major comments:

Figure 1(a) it is difficult to distinguish between the Rain-Wind and Wave-Wind. Instead of hatchings you may consider to use two colors and maybe try to zoom over the Mediterranean.

Figure 2 I suggest to express the compound frequencies as percentages (%). This will be easier to understand. If I understood correctly, you can simply multiply the current values by 100, in order to get the % of days with compound extremes within each season. Same for Figures 3,4,5,7

Figure 4(d)-(f). Please consider doing a statistical test to assess whether the frequency of compound extremes during cyclones is significantly different that the frequency of compound extremes when cyclones do not occur. You can try for example a proportion test. Same for Figure 7(d)-(f).

Minor comments:

L7 "weather compounds" is not clear.

L15 "of wave-wind extremes" not clear. Maybe "and wave-wind extremes".

L22 Please don't cite Wikipedia. I suggest to look first for peer-reviewed papers and then online newspaper articles from for example BBC, CNN, the Guardian.

L25 "compound extreme event"

L31 you may consider to add https://rmets.onlinelibrary.wiley.com/doi/full/10.1002/qj.3757

L49 I prefer to say "to compound extremes" instead of "to weather compounding". The latter terminology sounds a bit awkward and does not strictly refer to extremes.

L66 "compounding" or "compound"?

L69-79 please move it before L63, so that the description of the Sections will be in sequential order. You can also consider to convert L63-68 into sentences, i.e. not questions.

L88-89 "Moreover, the results…". Here in the Methods, I would simply state that you tested other percentile thresholds, also without referring to the figures. Then in the Results section, after you presented the plots, you can state "The above results are not sensitive to the….etc" also referring to the SM figure.

L122 "1979 to 2020". Didn't you use "1980 to 2019"?

L133 "weather compounds" please amend with "compounds" or "compound extremes"

L138 "schematic" or "scheme"?

L141-151 introduce the dynamical features in the same order as L137. Or vice versa.

L152 same as above.

L187-191 move to Results or Discussion section.

L211 "(Fig. 2)". Remove it

L213 "compounds" and add full stop.

L222 "weather compounds" change here and all over the text as mentioned before

---

## Author Comment (AC1)

We thank Paolo De Luca for his constructive comments on the manuscript. We will include his suggestions in the main manuscript as detailed in the line-by-line answer below. Note that the reviewer's comments are in **black text**, our answers in **red text**.

**REVIEWER 1: Paolo De Luca**
The paper by Portal et al. addresses an important thematic that in general deserves more attention by the community, which is the identification of dynamical drivers of compound extremes. In the paper, the authors focused on Mediterranean cyclones and compound rain-wind, wave-wind extremes, within the reanalysis period 1980-2019. They introduce a new definition of cyclone's impact area and assess how three different dynamical features contribute to the occurrence of the extremes. Then, they also quantify the most frequent cyclone type linked to the extremes and provide a discussion of their results.
The paper is sound, well written and therefore suitable for publication in WCD. Below you find some major and minor comments that I hope the authors will address in the review process.

Major comments:
- Figure 1(a) it is difficult to distinguish between the Rain-Wind and Wave-Wind. Instead of hatchings you may consider to use two colors and maybe try to zoom over the Mediterranean.
  We include below (and in the manuscript) a new version of Figure 1. In panel (a) we have zoomed closer to the cyclone and we have changed the visualisation of the extreme patterns, hoping this can also facilitate the visual identification of the compounds.

[Figure]

- Figure 2 I suggest to express the compound frequencies as percentages (%). This will be easier to understand. If I understood correctly, you can simply multiply the current values by 100, in order to get the % of days with compound extremes within each season. Same for Figures 3,4,5,7.
  We will change the notation from ratio to percentage.
- Figure 4(d)-(f). Please consider doing a statistical test to assess whether the frequency of compound extremes during cyclones is significantly different that the frequency of compound extremes when cyclones do not occur. You can try for example a proportion test. Same for Figure 7(d)-(f).
  Although we cannot yet show the results due to time constraints, we are willing to compute the test and to include the results in the new version of the manuscript.

Minor comments:
L7 "weather compounds" is not clear.

Has been corrected with « compound extremes ».

L15 "of wave-wind extremes" not clear. Maybe "and wave-wind extremes".

Has been corrected with « and of », where the preposition « of » refers to high incidence.

L22 Please don't cite Wikipedia. I suggest to look first for peer-reviewed papers and then online newspaper articles from for example BBC, CNN, the Guardian.

We have replaced the Wikipedia link with two online articles from DWD monthly weather report and from The Washington Post.

L25 "compound extreme event"

Has been corrected.

L31 you may consider to add https://rmets.onlinelibrary.wiley.com/doi/full/10.1002/qj.3757

We added the reference as suggested.

L49 I prefer to say "to compound extremes" instead of "to weather compounding". The latter terminology sounds a bit awkward and does not strictly refer to extremes.

We thank the reviewer for the useful correction, and have corrected the text.

L66 "compounding" or "compound"?

Has been corrected.

L69-79 please move it before L63, so that the description of the Sections will be in sequential order. You can also consider to convert L63-68 into sentences, i.e. not questions.

Although we understand the reason for the reviewer's suggestion, we prefer to maintain this order in the text because we believe that the present flow highlights the main research questions, rather than the more subtle methodological question on compound attribution.

L88-89 "Moreover, the results…". Here in the Methods, I would simply state that you tested other percentile thresholds, also without referring to the figures. Then in the Results section, after you presented the plots, you can state "The above results are not sensitive to the….etc" also referring to the SM figure.

We have modified the text as suggested.

L122 "1979 to 2020". Didn't you use "1980 to 2019"?

We have corrected giving the number of cyclones in the period 1980 to 2019.

L133 "weather compounds" please amend with "compounds" or "compound extremes"

Has been corrected.

L138 "schematic" or "scheme"?

We have replaced with « scheme ».

L141-151 introduce the dynamical features in the same order as L137. Or vice versa.

We have reversed the order in L137.

L152 same as above.

The order is now consistent across the Section. We thank the reviewer for pointing this out.

L187-191 move to Results or Discussion section.

We have shifted and adapted the paragraph to the Discussion section.

L211 "(Fig. 2)". Remove it

Has been removed.

L213 "compounds" and add full stop.

We have specified « maximum compound frequency » in the line above, although we are not sure this addresses the reviewer's correction.

L222 "weather compounds" change here and all over the text as mentioned before

This has been amended throughout the text.

---

## Author Comment (AC2)

We thank Reviewer #2 for his useful and detailed comments on the manuscript. We will include his suggestions in the main manuscript as described in the line-by-line answer below. Note that the reviewer's comments are in **black text**, our answers in **red text**.

**REVIEWER 2**
The authors examine the role of Mediterranean cyclones in the formation of compound extremes from different perspectives including the distance from the event, the dynamic features, and the type of the cyclone.

The paper is beautifully written with a comprehensive discussion of the results that is extensively supported by the literature. I consider that it should be accepted for publication, but there are some points that needs to be corrected or clarified.

Specific comments :

- Section 2.1a: Please justify why a coarser temporal and spatial resolution were chosen instead of the available ones from ERA5? The higher resolution wouldn't provide an added-value in the analysis performed?
  A few reasons, of conceptual and practical origin, underlie our choice of resolution.
  (i) Because we study the statistical relations between Mediterranean cyclones and extremes, the sample of extremes of interest is one where the spatial extent of the events is captured by both the low (0.5 deg) and by the high (0.25 deg) resolution dataset. In fact, we expect extreme impacts by synoptic weather systems to have a spatial extent greater than 0.5 deg, and to be well identified independently of the specific .5 / .25 deg resolution. See e.g. the example in Figure 1.
  (ii) A higher resolution dataset could be certainly useful for analysing (a) the characteristics of individual extreme events, in order to detail the interactions between atmospheric flow and topography, or (b) extremes of convective nature. However, (a) we expect climatological statistics to average out any small spatial details of the first kind, and (b), because ERA5 (even at 0.25 deg) does not resolve small-scale convective processes and topographic features, we do not focus on this type of extremes. We point out that, even using a lower 0.5 deg resolution, the climatological role of large-scale topographic features emerges naturally in our results with a sufficient pattern definition for the objective region at study  (e.g., see Figure 6, 4a-c).
  (iii) Limitations in ERA5 data storage capacity.
  We nonetheless acknowledge the reviewer's concern by mentioning the scale of the extreme events sampled by the dataset's resolution in the Methods (Section 2.1), where we specify:
  « A horizontal resolution of 0.5 deg suffices for the identification of surface extremes with scales from the order of 100~km, as those induced by synoptic-scale weather systems, but is unable to capture small-scale (convective-driven) extremes. »

- Section 2.1b: As also stated by the authors, negative wave biases are known especially for coastal areas. Why didn't the authors consider using a much higher resolution dataset for the waves available by Copernicus? I'm afraid that this

underestimation could largely affect the frequencies of the wave-wind compounds given that a threshold for wave (i.e., 2m) is set.

In the figure below we compare the 0.5 deg interpolation of wave-and-swell-height with the 0.25 deg dataset for the 11th of November at 12 CET (the same date and time as Figure 1). The cyclone centre is represented by the dark-red dot close to the Balearic Islands. From this case we observe some small changes in the field, but the intensity, location and extent of the wave-height maxima are consistent between the two maps. Hence, we expect small resolution-related changes in extreme wave events to be smoothed out from considering a large sample of cases.

[Figure]

Regarding the swell-and-wave-height systematic bias we point out that, although the data is affected by an underestimation of wave height, the bias is present independently of the presence / absence of a cyclone, hence is unlikely to affect our statistics based on event frequency - and quite independent of the absolute values. This is to say that the number and selection (i.e. the ordering) of the 2% most extreme events should be unaffected compared with an unbiased (or bias-corrected) dataset.

The 2 m minimum threshold for the selection of the extremes is data dependent, meaning that it is sensitive to the absolute values of the extremes in the specific dataset. The specific value was chosen in order to obtain a sample of wave extremes of consistent intensity across the Mediterranean region, i.e. by limiting the selection of extremes in regions where the 98th percentile of the distribution was substantially weaker than elsewhere.

Finally, the choice of resolution still complies with the reasoning detailed in the previous answer.

- Section 2.4 and in the rest of the text: *p(e)* is not absolute frequency, it's relative. The text has been corrected throughout.
- Unfortunately, almost all figures are difficult to read. I would suggest trying different colour palettes and patterns and/or increase the size. To solve this issue, in the new Figures we will increase the size of the panels. We hope this improves the graphical visualisation and we encourage the reviewer to point out specific colour schemes that they find difficult to read.

- I would suggest a proof reading of the text, as there are some errors regarding syntax and, mainly, punctuation.

  We will address the reviewer's concern during elaboration of the new version of the manuscript.

Minor comments:

- Abstract, line 11: What are these peaks? Spatial? Please clarify.
  Spatial peaks, specification inserted in text.
- Abstract, line 11: The "proportion of cyclone-related compounds" over what?
  We have rephrased to « The fraction of compounds happening within a cyclone's impact area... ». We hope it is clear now that we refer to the fraction of cyclone-related compounds over all compounds.
- Abstract in general: I think the abstract does not capture well the content of the paper.
  Although the comment is rather vague, we have tried to edit the abstract in order to better adapt it to the content and findings of the paper, as follows.
  « Mediterranean cyclones are the primary driver of many types of surface weather extremes in the Mediterranean region, the association with extreme rainfall being the most established. The large-scale characteristics of a Mediterranean cyclone, the properties of the associated airflows and temperature fronts, the interaction with the Mediterranean Sea and with the topography around the basin, and the season of occurrence, all contribute in determining its surface impacts. Here, we take these factors into account to interpret the statistical links between Mediterranean cyclones and compound extremes of two types, namely co-occurring rain--wind and wave--wind extremes. Compound extremes are attributed to a cyclone if they fall within a specially defined *Mediterranean cyclone impact area*. Our results show that the majority of Mediterranean rain--wind and wave--wind extremes occur in the neighbourhood of a Mediterranean cyclone, with local peaks exceeding 80%. The fraction of compounds happening within a cyclone's impact area is highest when considering transition seasons, and for rain--wind events compared with wave--wind events. Winter cyclones, matching with the peak occurrence of large and distinctively baroclinic cyclones, are associated with the highest compound frequency. A novelty of this work, the de-construction of cyclones' impact areas based on the presence of objectively-identified air streams and fronts, reveals a high incidence of both types of compound extremes below warm conveyor belt ascent regions, and of wave--wind extremes below regions of dry intrusion outflow. »
- Line 53: What do you mean with "objective regions"?
  We have replaced this formulation by « at a global scale ».
- Section 2.2.1: Consider adding a table that could be used as a reference for section 4.3, containing the number of the cluster, the season in which appears and the associated weather configuration (and maybe the region of max occurrence).
  We are thankful for the suggestion and will add such a table in the new manuscript.

- Line 152: Could you elaborate on how the "unit valued grid points" are defined?
  Unit-valued grid points are defined based on the fulfilment of the individual feature conditions listed in the lines preceding 152. Extra clarifications on how the boolean feature masks are defined will be provided within the paragraph in the new version of the manuscript.
- Line 222: Which these factors are?
  These factors are mainly geographical exposure and seasonal changes in the local climatological weather conditions. We have expanded the text to include this.
- Lines 234-236: Could you elaborate on why this happens?

  Following the reviewer's suggestion, we have detached a new paragraph to elaborate more on the peculiarities in the statistics of the south-eastern Mediterranean region. The paragraph reads as below.

  « The occurrence of winter R∧W events outside cyclones' impact areas is particularly high in the eastern and southern sectors of the Mediterranean (Fig. 4(h)). Although the fraction of compound events associated with cold fronts and warm conveyor belts peaks in the central Mediterranean (Fig. 5), during the south-eastern Mediterranean winter we detect a larger fraction of stray features amongst those co-occurring with R∧W compounds than elsewhere (Fig. 5(b),(e), where hatched areas indicate that more than 90% of the features occurring with compounds are attached to a cyclone). The relatively low local extremal thresholds (Fig. SM1), implying a selection of weather extremes of weak-to-moderate intensity, may be at the origin of the low fractions of cyclone-related compounds and of the higher relevance of stray dynamical features in the region. »

Very minor comments :

- Lines 18 & 21: 5 November or November 5$^{th}$ or the 5$^{th}$ of November
  Modified according to the reviewer's advice.
- Line 22: I don't think Wikipedia is an appropriate citation. Please consider other sources.
  We have replaced the Wikipedia link with two online articles from DWD monthly weather report and from The Washington Post.
- Lines 52-55: Please check syntax
  We have changed the lines as follows.
  « Compounding of rain–wind and wave–wind extremes and its relation to extratropical cyclones has received attention at a global scale (Owen et al., 2021; Ridder et al., 2020; Catto and Dowdy, 2021). Although studies on Mediterranean rain–wind extremes exist, these consider a small sample of large-scale events (Raveh-Rubin and Wernli, 2015) or take a Lagrangian cyclone-centred perspective (Rousseau-Rizzi et al., 2023). »
- Line 157: n is not defined; I suppose is the number of events. And N equals ~7200?
  We confirm that n is the number of events, and we have inserted the definition in the text.
- Lines 175-177: Check syntax

- To improve readability, we have rephrased the whole paragraph as follows.
  « Mediterranean cyclones are on average weaker, smaller and shorter-lived than their North-Atlantic equivalents (Trigo et al., 1999; Trigo, 2006; Čampa and Wernli, 2012; Campins et al., 2011). However, a Mediterranean cyclone's impact area must account for the interactions of the induced atmospheric flow with coastal boundaries and orographic features, since these modify the spatial distribution of the surface impacts (Pfahl, 2014; Houze Jr, 2012; Obermann-Hellhund, 2022; Owen et al., 2021; Flaounas et al., 2019) compared with a conceptual airflow model of an extratropical cyclone over the ocean (Carlson, 1980; Wernli and Davies, 1997; Schultz, 2001). The position of a cyclone centre relative to the coast and to the Mediterranean sea, the main source of moisture, also determines the character of the impacts (Jansa et al., 2001; Pfahl, 2014). »
- Line 179: Which year?
  Year 1980. We thank the reviewer for noting this and have completed the sentence.
- Caption Figure 2: I don't see "negative dashed" red contours. Is this correct?
  We have corrected the adjectivation, now « negative » refers to the difference.
- Line 211: "most" instead of "must"
  Mistake corrected.
- Line 220: "towards the south west" in which season?
  In spring. We are grateful to the reviewer for signalling our oversight.
- Line 314: "Lions" instead of "Lion"
  We looked this up and both spellings seem to be correct in English.
- Lines 471-472: This paper is published.
  Reference corrected.
- Lines 494-495: This paper is published (2024)
  Reference corrected.
- Line 518: Journal and pages are missing
  This is a book. We have added the total number of pages.
- Line 559-560: This is a pre-print now
  Reference corrected.
- Line 567: Journal and pages are missing

  Reference corrected.

---

## Author Response (AR2)

We thank the editor for the careful reading of the manuscript and for the constructive comments. Please note that the editor's comments are reported in **black text**, our answers in **red text**.

**CO-EDITOR : Irina Rudeva**

I appreciate the changes made to the manuscript and will ask the reviewers to further assess those changes.

Additionally, I would like to clarify the usage of the term 'cut-off lows'. E.g., line 145 in the version with tracked changes says: "clusters 6 and 9, often land based, are shallow summer lows (Sharav heat low and short-wave cut-off)'. What do you mean by shallow cut-off lows? Are they confined to the upper troposphere? Based on Fig 3 Givon et al 2021, I agree that cluster 6 is a shallow (low-level) cyclone but, e.g., cluster 8 also shows a signature of a cut-off low; furthermore, RWB in clusters 2 and 5 suggests a formation cut-off low.

We thank the editor for their accurate review, and hope to clarify with the following.

The correct paper describing the clusters is Givon et al. 2024, available at this link.

In line 145, we refer to shallow lows as to low-pressure systems which have a distinct footprint in the lower troposphere and a less distinct pattern in the upper levels (cf. mslp and PV patterns in Givon et al. 2024's Figure 3). To clarify in the text, we specify that « clusters 6 and 9, often land based and displaying a shallow low-level structure, are summer lows ».

Regarding cluster names, these were justified by checking PV cut-off and streamer masks (Appendix C in Givon et al. 2024) and other dynamical features such as cyclone mobility. The specific cyclone clusters displaying cut-off low signatures are discussed in the points below :

- Clusters 5 and 2. While it is true that anti-cyclonic wave breaking often terminates with cut-offs in various levels, these clusters are dominated by a PV streamer that is still connected to the main PV reservoir at the time of the cyclones' maximum intensity (see shading in Givon et al. 2024's Figure 3). In fact, these cyclones tend to peak during their wave breaking phase, before a possible formation of a cut-off.

- Cluster 8, similarly to clusters 2 and 5, shows a PV streamer that is connected with the main PV reservoir. Cyclonic wave breaking is also known to produce cut-off lows, but these tend to get refracted poleward and merge back with the PV reservoir.

To conclude, the cluster names chosen in Givon et al. 2024 are useful to relate to the cluster's dominant driving mechanism, yet do not exclude a-priori the formation of cut-off lows at different stages of the cyclones' development. Here we prefer to relate to the same names for consistency with the reference study.